

# Simulating thermal density operators
# with cluster expansions and tensor networks

**Bram Vanhecke[1,2], David Devoogdt[1], Frank Verstraete[1] and Laurens Vanderstraeten[1]**

**1** Department of Physics and Astronomy, University of Ghent,
Krijgslaan 281, 9000 Gent, Belgium
**2** University of Vienna, Faculty of Physics, Boltzmanngasse 5, 1090 Wien, Austria

## Abstract

We provide an efficient approximation for the exponential of a local operator in quantum spin systems using tensor-network representations of a cluster expansion. We benchmark this cluster tensor network operator (cluster TNO) for one-dimensional systems, and show that the approximation works well for large real- or imaginary-time steps. We use this formalism for representing the thermal density operator of a two-dimensional quantum spin system at a certain temperature as a single cluster TNO, which we can then contract by standard contraction methods for two-dimensional tensor networks. We apply this approach to the thermal phase transition of the transverse-field Ising model on the square lattice, and we find through a scaling analysis that the cluster-TNO approximation gives rise to a continuous phase transition in the correct universality class; by increasing the order of the cluster expansion we find good values of the critical point up to surprisingly low temperatures.



# 1 Introduction

In quantum-many body systems, the exponential of the many-body Hamiltonian $H$ often plays a fundamental role. Indeed, for quantum systems at finite temperature, the thermal density operator

$$\rho(\beta) = \frac{1}{\mathcal{Z}(\beta)} e^{-\beta H}, \qquad \mathcal{Z}(\beta) = \text{Tr}\left(e^{-\beta H}\right), \tag{1}$$

contains all static information, whereas the time evolution of a quantum state is dictated by the time-evolution operator

$$U(t) = e^{-iHt}. \tag{2}$$

Efficient numerical schemes for representing such exponentials are therefore crucial for simulating many-body systems. The most common approach is the use of a Trotter-Suzuki expansion [1,2], which breaks up the exponential operator into a sequence of local gates. This approach is size-extensive, a crucial property for simulating uniform systems directly in the thermodynamic limit [3]. As a downside, the Trotter-Suzuki expansion breaks translation symmetry and is necessarily limited to local interactions and small steps in (real or imaginary) time. For one-dimensional (1-D) systems an alternative approach [4] uses the formalism of matrix product operators [5], which preserves all symmetries, is size-extensive and works for long-range interactions; here, the downside is that the MPO is correct only up to first order, and going to higher orders is not straightforward. Finally, in the context of quantum Monte-Carlo simulations, the use of series expansions [6,7] has proven very useful, but such an approach is not size-extensive.

In Ref. [8], it was realized how a series expansion can be encoded in the language of tensor networks in a way that *is* size-extensive and can, therefore, be naturally formulated directly in the thermodynamic limit. Motivated by the formal results on the representability of thermal states as tensor network operators [9–11], the tensor-network construction was recently improved [12] by considering clusters instead of the bare terms in the series expansion. Here, a cluster is essentially a regrouping of many different terms that act non-trivially on a small patch of the lattice, including many higher-order terms that a truncated series expansion would neglect. In Ref. [12], it was indeed realized that such a cluster expansion can be encoded as a tensor network operator (TNO) with moderate bond dimension, in a way that is size-extensive, preserves all spatial and internal symmetries and works in any dimension. It was shown that such a "cluster TNO" is a very efficient numerical tool for (i) simulating simulating the real-time evolution of a global quench in a 1-D spin chain, or (ii) optimizing a ground-state approximation with a projected entangled-pair state. In both cases, the ability to take large real- or imaginary-time steps proved to be a very efficient feature of the cluster-TNO approach.

Motivated by these results, in this paper we use the cluster-TNO approach for simulating thermal density operators of two-dimensional quantum spin systems. In Sec. 2, we first reit-

erate the general idea of an extensive cluster expansion, and how tensor networks provide a natural expression. We further elaborate on different constructions in one and two dimensions. In Sec. 3, we benchmark these different constructions by comparing to the exact exponential on finite clusters, by checking to what extent a cluster-TNO approximation for the real-time evolution operator is a unitary operator and how a cluster TNO allows us to compute the density of states. Finally, in Sec. 4 we apply cluster TNOs to simulate the thermal phase transition of the quantum transverse-field Ising model in two dimensions.

## 2 Construction

### 2.1 General idea

Let us first explain the general idea of a cluster expansion, and how we can encode this efficiently as a tensor network operator. We consider a completely general lattice with spins on every site, directly in the thermodynamic limit, and a translation-invariant operator that is the sum of local terms,

$$H = \sum_n h_n \,, \tag{3}$$

where $h_n$ is a local operator that only acts on finite region $n$. The exponential of this operator can be written down as a series expansion of the form

$$T = \exp\left(\lambda \sum_n h_n\right) = \sum_{p=0}^{\infty} \frac{\lambda^p}{p!} \left(\sum_n h_n\right)^p \,, \tag{4}$$

where $\lambda$ is thought to be a small parameter. In such a series expansion, the different terms in $p$-th order are not extensive: when applying the $p$-th order term to a uniform state, the state is no longer normalizable in the thermodynamic limit. Therefore, we propose a specific regrouping of the terms in the series expansion,

$$T = \sum_{n=1}^{\infty} \mathcal{T}_n \,, \tag{5}$$

where the $\mathcal{T}_n$ contains *all* terms in the expansion in Eq. (4) that have maximum cluster size $n$. Here, the maximum cluster size of a given term in the series expansion is the largest region of the lattice on which there are operators acting non-trivially, and which cannot be decomposed as a tensor product of smaller clusters. So, each $\mathcal{T}_n$ contains terms in the series expansion of all orders. Below, using tensor networks we will indicate how a truncated series expansion indeed leads to a size-extensive operator.[1]

Let us make things concrete for a 1-D chain. A one-site cluster $\mathcal{S}_1$ is of the form

$$\mathcal{S}_1 = \boxed{\diagup} = \mathcal{O}_1 - \mathbb{1} \,, \tag{6}$$

and contains all terms in the exponentiated Hamiltonian $T$ [Eq. (4)] that act non-trivially on a single site. $\mathcal{O}_1$ is thus the exponentiated Hamiltonian acting on a 1-site system; in general, we define $\mathcal{O}_n$ as the exponential of the Hamiltonian, restricted to a patch of $n$ sites. The two-site cluster $\mathcal{S}_2$ is given by taking all terms that act non-trivially on two sites, and subtracting all

---

[1]In this work, our discussion of a cluster expansions and its extensivity is situated on an intuitive and practical level, but we refer to the more formal results [9–11] for a more rigorous discussion.

terms that can be decomposed into one-site clusters,

$$\mathcal{O}_2 - \mathbb{1} = \mathcal{S}_2 + \mathcal{S}_1 \otimes \mathcal{S}_1 \tag{7}$$

$$= \;\boxed{\phantom{xx}} + \boxed{\phantom{x}}\boxed{\phantom{x}}\; . \tag{8}$$

Similarly, the three-site cluster $\mathcal{S}_3$ is given by summing all terms that act non-trivially on three sites and subtracting all contributions that can be decomposed into one- and two-site clusters,

$$\mathcal{O}_3 - \mathbb{1} = \mathcal{S}_3 + \mathcal{S}_2 \otimes \mathcal{S}_1 + \mathcal{S}_1 \otimes \mathcal{S}_2 + \mathcal{S}_1 \otimes \mathcal{S}_1 \otimes \mathcal{S}_1 \tag{9}$$

$$= \;\boxed{\phantom{xxx}} + \boxed{\phantom{xx}}\boxed{\phantom{x}}$$

$$+ \boxed{\phantom{x}}\boxed{\phantom{xx}} + \boxed{\phantom{x}}\boxed{\phantom{x}}\boxed{\phantom{x}}\; . \tag{10}$$

This procedure can be extended to increasing cluster size $m$: we compute all terms that act non-trivially on $m$ sites and we subtract all terms that can be decomposed into smaller clusters.

The $\mathcal{T}_n$ introduced in Eq. (5) is then made up of the superposition of all possible tensor products of $\mathcal{S}_m$ with $m \leq n$ and at least one cluster of size $n$.

This cluster expansion can be straightforwardly generalized to two dimensions. We take regions of increasing size, compute the non-trivial terms in $T$ on this region, and subtract all terms that can be decomposed into clusters of smaller size. As an example, the cluster on a two-by-two region is defined as

$$O_{2\times2} - \mathbb{1} = \;[\text{tensor network diagrams}] \;, \tag{11}$$

where the last operator is the four-site cluster we need, and all previous terms are decompositions into smaller clusters.

As such the cluster expansion is a formal tool for grouping terms that appear in $T$, but now we use tensor networks to represent such a cluster expansion in a natural way. Suppose we want to represent a term in the cluster expansion of the form

$$[\text{tensor network diagram}]. \tag{12}$$

We can encode such a configuration into a tensor network by associating to every site a six-leg tensor, with two legs that correspond to the physical action of the tensor network operator and four virtual legs that encode the cluster configuration:

$$ (13) $$

Here, the level '0' on the virtual legs is of dimension one, and is used between disconnected clusters. The higher virtual levels are used within the clusters, and arise from the tensor decomposition of the clusters – the specific method for finding these tensor entries will be the subject of the following subsections. The dimension of these higher virtual levels will generally be larger than one.

As such, the tensor network in Eq. (13) represents a single term in the cluster expansion. We can now sum up *all* terms in the cluster expansion by incorporating all tensor entries that appear in the above network into a single tensor, and repeating this tensor on every site in the lattice

$$ (14) $$

The tensor network operator (TNO) that we construct in this way now represents the sum of all of the above configurations.

One important feature of this "cluster TNO" is its extensivity, which reproduces the extensivity of the exponential. Concretely, this means that if the cluster TNO contains a term with a certain cluster on region $A$ and a term with another cluster on a non-overlapping region $B$, it also contains the term with both non-overlapping clusters. This implies that the cluster-TNO with clusters up to a certain size contains all terms in $T$ with non-trivial clusters, including the terms that are direct products of clusters on non-overlapping regions.

## 2.2 One-dimensional models

This general idea of encoding a cluster expansion into a tensor-network operator is made clear by working out the case of a local translation-invariant Hamiltonian in one dimension. We want to approximate the exponentiated Hamiltonian by a matrix product operator (MPO), which we can represent directly in the thermodynamic limit as

$$ \exp\left(\lambda \sum_n h_n\right) \approx \quad \cdots -\!\!\oslash\!\!-\!\!\oslash\!\!-\!\!\oslash\!\!- \cdots \qquad (15) $$

It appears that a cluster TNO is not unique; here, we explain three different constructions.

### 2.2.1 Type A

In the type-A construction, the clusters are encoded in the MPO as follows. The one-site cluster is encoded as a simple on-site operator with the virtual level '0' on both sides,

$$\mathcal{S}_1 = \quad \oslash \quad + \mathbb{1} \,. \tag{16}$$

The virtual level '0' is not drawn in this and following figures. Next, we introduce a single virtual level '1' for encoding the two- and three-site clusters

$$\mathcal{S}_2 = \quad \oslash \!\!\overset{1}{\rule{1em}{0.4pt}}\!\! \oslash \quad , \tag{17}$$

$$\mathcal{S}_3 = \quad \oslash \!\!\overset{1}{\rule{1em}{0.4pt}}\!\! \oslash \!\!\overset{1}{\rule{1em}{0.4pt}}\!\! \oslash \quad . \tag{18}$$

Here, the tensor entries $0-1$ and $1-0$ are found by, e.g., performing a singular-value decomposition (SVD) of the two-site cluster $\mathcal{S}_2$. The $1-1$ entry is then found by solving a linear problem.[2] We go on to the four- and five-site clusters, for which we introduce a new virtual level '2',

$$\mathcal{S}_4 = \quad \oslash \!\!\overset{1}{\rule{1em}{0.4pt}}\!\! \oslash \!\!\overset{2}{\rule{1em}{0.4pt}}\!\! \oslash \!\!\overset{1}{\rule{1em}{0.4pt}}\!\! \oslash \quad , \tag{19}$$

$$\mathcal{S}_5 = \quad \oslash \!\!\overset{1}{\rule{1em}{0.4pt}}\!\! \oslash \!\!\overset{2}{\rule{1em}{0.4pt}}\!\! \oslash \!\!\overset{2}{\rule{1em}{0.4pt}}\!\! \oslash \!\!\overset{1}{\rule{1em}{0.4pt}}\!\! \oslash \quad . \tag{20}$$

Obviously, this construction can be continued to include larger clusters. The bond dimension of the virtual levels, however, increases exponentially with the cluster size: the levels '1' and '2' have a dimension of $d^2$ and $d^4$, resp., with $d$ physical dimension. For larger clusters, we can choose to lower the bond dimension by truncating the singular values.

One important feature of the type-A construction involves the diagonal entries such as the $1-1$ entry that we have included for the three-site cluster. Indeed, this entry does not only include the three-site cluster into the MPO, but also gives rise to longer strings of the form

$$\oslash \!\!\overset{1}{\rule{1em}{0.4pt}}\!\! \oslash \!\!\overset{1}{\rule{1em}{0.4pt}}\!\! \oslash \!\!\overset{1}{\rule{1em}{0.4pt}}\!\! \oslash \quad . \tag{21}$$

We can correct for this contribution by redefining the $1-2$ and $2-1$ entries as

$$\mathcal{S}_4 - \quad \oslash \!\!\overset{1}{\rule{1em}{0.4pt}}\!\! \oslash \!\!\overset{1}{\rule{1em}{0.4pt}}\!\! \oslash \!\!\overset{1}{\rule{1em}{0.4pt}}\!\! \oslash \quad = \quad \oslash \!\!\overset{1}{\rule{1em}{0.4pt}}\!\! \oslash \!\!\overset{2}{\rule{1em}{0.4pt}}\!\! \oslash \!\!\overset{1}{\rule{1em}{0.4pt}}\!\! \oslash \quad . \tag{22}$$

In general, we can correct for the longer strings in the definition of the next virtual levels. If the bond dimension of these next virtual levels becomes too high, we can no longer correct for the longer strings. It is, a priori, unclear what are the effects of these contributions on the accuracy of the cluster TNO.

---

[2]It is important for numerical stability to solve the linear problem imposed by, e.g., Eqs.(18) and (20) rather than inverting the tensor entries $0-1$ and $1-0$ directly. For the larger clusters, the inversion problem can be written as the inversion of a a direct product of matrices; here, for numerical stability it is advised to first perform a singular-value decomposition of each matrix separately, and constructing a suitable pseudoinverse, instead of solving the linear problem directly.

### 2.2.2 Type B

The MPO encoding can be adapted to avoid inclusion of these strings of diagonal entries. Type B has the following entries

$$\mathcal{S}_1 = \;\;\varnothing\;\; + \mathbb{1}\,, \tag{23}$$

$$\mathcal{S}_2 = \;\;\varnothing\!\!-\!\!\overset{1}{\phantom{x}}\!\!-\!\!\varnothing\;\;, \tag{24}$$

$$\mathcal{S}_3 = \;\;\varnothing\!\!-\!\!\overset{1}{\phantom{x}}\!\!-\!\!\varnothing\!\!-\!\!\overset{1'}{\phantom{x}}\!\!-\!\!\varnothing\;\;, \tag{25}$$

$$\mathcal{S}_4 = \;\;\varnothing\!\!-\!\!\overset{1}{\phantom{x}}\!\!-\!\!\varnothing\!\!-\!\!\overset{2}{\phantom{x}}\!\!-\!\!\varnothing\!\!-\!\!\overset{1'}{\phantom{x}}\!\!-\!\!\varnothing\;\;, \tag{26}$$

$$\mathcal{S}_5 = \;\;\varnothing\!\!-\!\!\overset{1}{\phantom{x}}\!\!-\!\!\varnothing\!\!-\!\!\overset{2}{\phantom{x}}\!\!-\!\!\varnothing\!\!-\!\!\overset{2'}{\phantom{x}}\!\!-\!\!\varnothing\!\!-\!\!\overset{1'}{\phantom{x}}\!\!-\!\!\varnothing\;\;, \tag{27}$$

where the primed levels are entirely new levels, thus avoiding any diagonal entries. The primed levels and unprimed levels can only meet in the middle of the patch, and hence longer chains such as eq. (21) are excluded by this encoding. This comes at the numerical cost of twice the total bond dimension in comparison to the type-A construction.

### 2.2.3 Type C

Both the type-A and type-B construction requires us to solve linear problems for finding some entries, which can become ill-conditioned. In order to avoid this issue, we propose the type-C construction

$$\mathcal{S}_1 = \;\;\varnothing\;\; + \mathbb{1}\,, \tag{28}$$

$$\mathcal{S}_2 = \;\;\varnothing\!\!-\!\!\overset{1'}{\phantom{x}}\!\!-\!\!\varnothing\;\; + \;\;\varnothing\!\!-\!\!\overset{1}{\phantom{x}}\!\!-\!\!\varnothing\;\;, \tag{29}$$

$$\mathcal{S}_3 = \;\;\varnothing\!\!-\!\!\overset{1}{\phantom{x}}\!\!-\!\!\varnothing\!\!-\!\!\overset{1}{\phantom{x}}\!\!-\!\!\varnothing\;\;, \tag{30}$$

$$\mathcal{S}_4 = \;\;\varnothing\!\!-\!\!\overset{1}{\phantom{x}}\!\!-\!\!\varnothing\!\!-\!\!\overset{2}{\phantom{x}}\!\!-\!\!\varnothing\!\!-\!\!\overset{1}{\phantom{x}}\!\!-\!\!\varnothing\;\; + \;\;\varnothing\!\!-\!\!\overset{1}{\phantom{x}}\!\!-\!\!\varnothing\!\!-\!\!\overset{2'}{\phantom{x}}\!\!-\!\!\varnothing\!\!-\!\!\overset{1}{\phantom{x}}\!\!-\!\!\varnothing\;\;, \tag{31}$$

$$\mathcal{S}_5 = \;\;\varnothing\!\!-\!\!\overset{1}{\phantom{x}}\!\!-\!\!\varnothing\!\!-\!\!\overset{2}{\phantom{x}}\!\!-\!\!\varnothing\!\!-\!\!\overset{2}{\phantom{x}}\!\!-\!\!\varnothing\!\!-\!\!\overset{1}{\phantom{x}}\!\!-\!\!\varnothing\;\;. \tag{32}$$

The unprimed entries that we have introduced are arbitrary unitary tensors (up to a constant factor), and form the least squares solution to the problem, whereas the primed entries make sure we reproduce the clusters of even size. The ill-conditioning of the linear problems is now avoided since it reduces to inverting these unitary tensors. The downside is that the type-C constructions requires twice the bond dimension of the type-A construction.

Of course, the type-C construction can be combined with the type-A or type-B one: we can switch to the type-C prescription for the larger clusters, whenever the linear problem becomes ill-conditioned.

## 2.3 Two-dimensional models

For the 2-D case, we can make a distinction between two types of clusters: linear clusters (including branchings) and loops. For the former, we can straightforwardly extend the 1-D constructions, but the latter requires extra ingredients for representing them in terms of TNOs. In the following, we will consider the square lattice only, but our discussion also applies to other 2-D lattices

### 2.3.1  Linear clusters and branchings

The one and two-site clusters are simply encoded as

$$\tag{33}$$

where these entries are again found by taking singular-value decompositions of the two-site clusters. There are six different three-site clusters,

$$\tag{34}$$

and their rotations; these diagonal entries are, again, found by solving simple linear problems. We can add larger clusters of the form

$$\tag{35}$$

without increasing the bond dimension. But if we want to include still larger clusters such as

$$\tag{36}$$

we need to include extra virtual levels. Clearly, we can again continue this construction to include larger and larger clusters. We have to take care that for before including a new cluster, we have included all smaller clusters that fit within the new one.

Here, we have chosen the type-A construction with diagonal TNO entries, that give rise to longer strings in the TNO. We could avoid these longer strings by resorting to the 2-D version of the type-B and type-C constructions.

### 2.3.2  Loops

Starting with the two-by-two cluster, we can also have clusters that contain loops; for these clusters, we cannot simply perform the simple growing of the TNO as we did for the 1-D case. Instead, we need to introduce a new virtual level, such that we can represent the two-by-two cluster

$$\tag{37}$$

where we use Greek letters for labeling the virtual levels that give rise to loops. Finding these entries can be done by a sweeping algorithm, similar to a variational optimization of a periodic matrix product state [13]. Additionally, we can add linear parts to these loop clusters, such as

$$\tag{38}$$

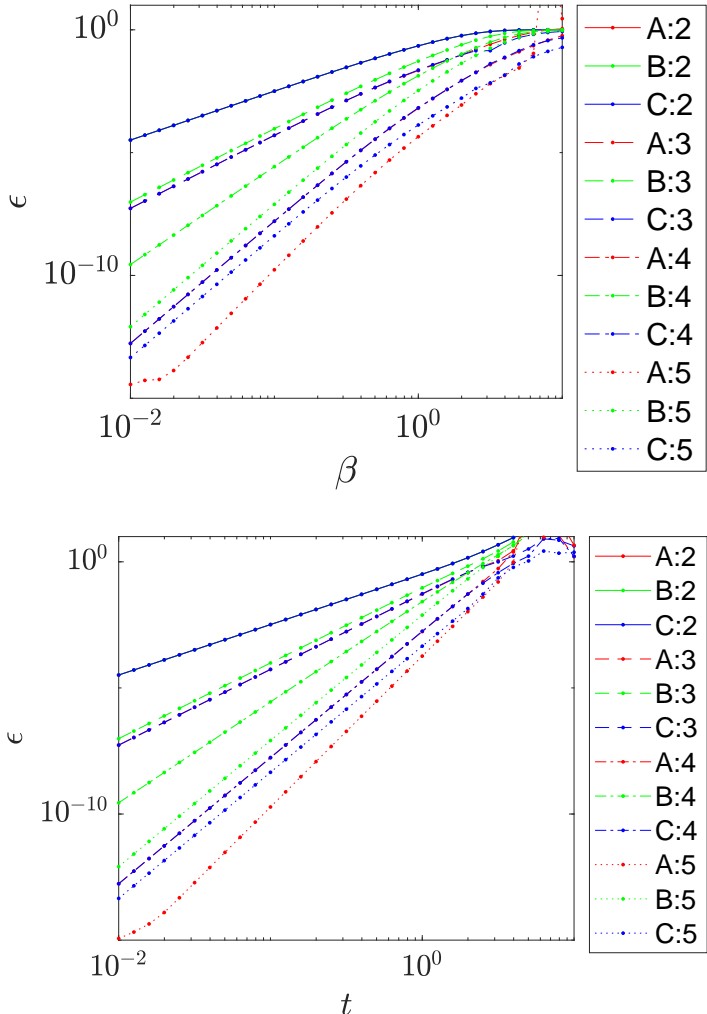

Figure 1: Relative error $\epsilon$ of the cluster-MPO for the 1-D spin-1/2 Heisenberg model on a periodic chain of 11 sites. Top panel shows $\epsilon$ for the unnormalized thermal density operator $\rho = \mathrm{e}^{-\beta H}$, as a function of inverse temperature $\beta$. Bottom panel shows $\epsilon$ of the real-time evolution operator $U(t) = \mathrm{e}^{-iHt}$, as a function of time $t$. We show results of the cluster-MPO of types A,B and C with cluster sizes $c = 2, 3, 4, 5$.

## 3 Benchmarks for 1-D models

In this section, we investigate how accurate these cluster TNOs are for representing exponentials of nearest-neighbour spin-chain Hamiltonians. In particular, we will compare the different types of MPO constructions that we have introduced in the previous section.

### 3.1 Accuracy on finite chains

As a first benchmark, we compare the cluster TNO with the exact matrix exponential on a finite periodic system. We will compute the relative 2-norm error $\epsilon$

$$\epsilon = \frac{||U_{\text{exact}} - U_{\text{TNO}}||_2}{||U_{\text{exact}}||_2}. \tag{39}$$

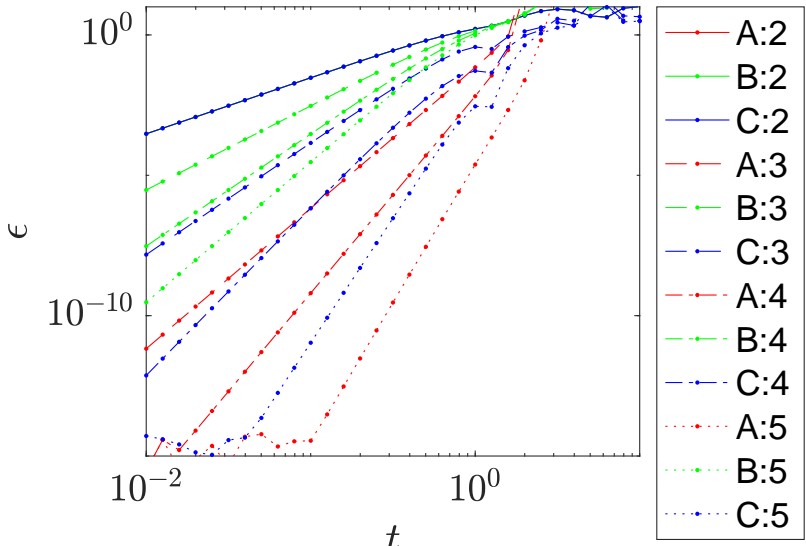

Figure 2: Relative error $\epsilon$ of the cluster-expansion TNO for the real-time evolution operator $U(t) = e^{-iHt}$ of the 1-D transverse-field Ising model at $g = 1$ on a periodic chain of 11 sites. We show results of the cluster-expansion TNO of type A,B and C with cluster size $c = 2, 3, 4, 5$, as a function of time $t$.

In Fig. 1, we first consider the spin-1/2 Heisenberg Hamiltonian,

$$H = \sum_i \sigma_i^x \sigma_{i+1}^x + \sigma_i^y \sigma_{i+1}^y + \sigma_i^z \sigma_{i+1}^z, \tag{40}$$

and show the accuracy of the cluster MPO for both the thermal density operator at inverse temperature $\beta$ and the real-time evolution operator as a function of time $t$. Clearly, all the expansions improve when the order is increased. One would expect the type-B construction to be the better one of the three, as it avoids the presence of longer strings in the MPO. Evidently, this is not the case for the two examples that we consider. This implies that the terms corresponding to the longer strings provide an approximation of the larger clusters. Moreover, type A outperforms type C by quite some margin for sufficient low temperatures. As the type-A construction also has a lower bond dimension, this is clearly the better choice. For completeness, in Fig. 2 we also show the accuracy of the cluster MPO for the transverse-field Ising model with Hamiltonian

$$H = -\sum_i \sigma_i^x \sigma_{i+1}^x + g \sigma_i^z, \tag{41}$$

with similar results as for the Heisenberg model.

## 3.2 Unitarity of the cluster expansion

One could wonder to what extent the cluster expansion represents a unitary operator. To assess this, we calculate the 1-site reduced density matrix of the operator $\rho = UU^{\dagger} = e^{-iHt} e^{-iHt} \approx I_d$ directly in the thermodynamic limit. The unitarity error $\epsilon$ is defined as

$$\epsilon = ||\rho - I_d||_2. \tag{42}$$

The results for the Heisenberg model are shown in Fig. 3. The expansions become in general more unitary with increasing cluster expansion order. Once again, type-A outperforms the others by quite some margin and scales better with cluster-expansion order.

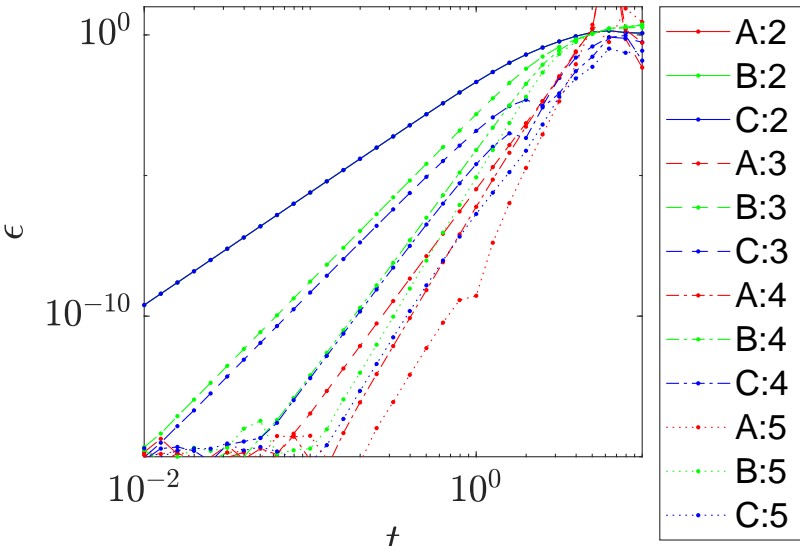

Figure 3: Unitarity error $\epsilon$ as a function of time t for the 1-D spin-1/2 Heisenberg model. We show results of the cluster-expansion TNO of type A,B and C with clustersize c=2,3,4,5.

## 3.3 Spectral Energy Density

As a final benchmark, we will consider the spectral energy density. For a generic spin Hamiltonian $H$ of $N$ sites with local dimension $d$, this quantity is defined as

$$\mu(\omega) = \frac{1}{d^N} \sum_j \delta(E_j - x), \tag{43}$$

where $E_j$ are the eigenvalues of $H$. It can be computed directly from the trace of the time-evolution operator [14] as

$$\mu(\omega) = \frac{1}{d^N} \int \frac{dt}{2\pi} e^{-i\omega t} U(t), \quad U(t) = \text{Tr}\left(e^{iHt}\right). \tag{44}$$

Using the cluster expansion, we can obtain an efficient tensor-network representation of $U(t)$ up to a certain time $T$. In order to avoid cutting of the approximate $U(t)$ too sharply, we multiply it with a Gaussian window function

$$\tilde{\mu}(\omega) = \frac{1}{d^N} \int \frac{dt}{2\pi} e^{-i\omega t} U(t) e^{-\alpha \frac{t^2}{2T^2}}, \tag{45}$$

resulting in a smeared-out spectral density function $\tilde{\mu}$ with a resolution $\mathcal{O}(T^{-1})$.

We consider the 1D Ising model with transverse field $g = 1$ and longitudinal field $h = 1$

$$H = -\sum_i \sigma_i^x \sigma_{i+1}^x + g\sigma_i^z + h\sigma_i^x, \tag{46}$$

on a system of 10 sites with periodic boundary conditions; we present the results in Fig. 4. In the top panel, we have plotted the accuracy of the cluster-expansion approximation for $U(t)$, by comparing it to the exact result. We observe that the cluster expansion is a good approximation for $t \lesssim 3$. In the bottom panel, we show the spectral densities, convoluted with a Gaussian. We observe that the cluster-TNO provides an accurate simulation of the density of states, up to the fine-grained features that require longer times in $U(t)$.

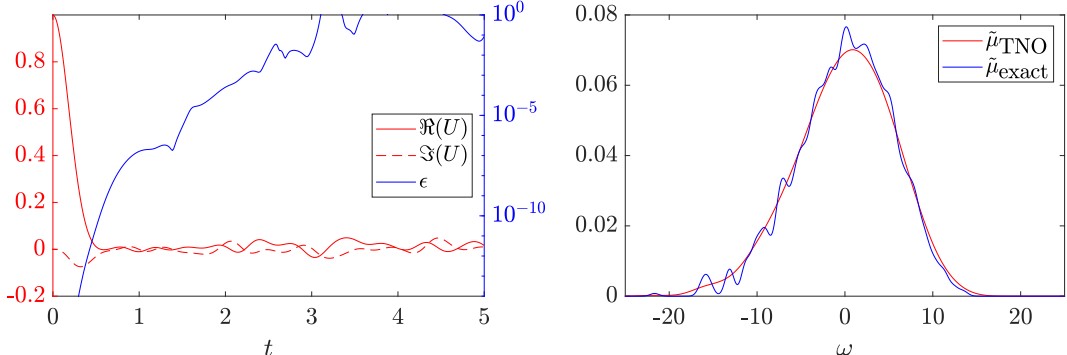

Figure 4: Results for the density of states of the longitudinal-field Ising model [Eq. (46)] with $g = 1$ and $h = 1$ on a ring of 10 sites. In the left panel, the red lines show the real and imaginary parts of $U(t) = \text{Tr}(e^{-i\hat{H}t})$; the blue line shows the deviation between the exact result and the TNO. The right panel shows the smeared $\tilde{\mu}(\omega)$ calculated exactly ($T = 2$) and with TNO ($T = 0.6$).

### 3.4 Discussion

We have found that the cluster expansion TNO indeed approximates the Hamiltonian exponentials well. Surprisingly, we have also found that the type-A construction outperforms the others. This implies that the -unwanted- strings of 1-1 and 2-2 discussed in Section. 2 are actually lowering the error. Note, however, that this is most likely a feature only found for nearest neighbour interacting models, where the 1-1 strings are suppressed by a power in $t/\beta$. This would not be the case for further neighbour interactions. We therefore anticipate type-B to be superior, if not necessary, for such models.

Note also that while the error goes down when including higher order clusters, it always tends to saturate around $t/\beta \sim O(1)$, as can be clearly seen in Fig. 1,2, and 3, which is not unreasonable since the cluster expansion is also an expansion in powers of $t/\beta$. This should be taken into account when attempting to use this construction for low temperatures or long times.

## 4 Application: thermal density operators

Let us now consider two-dimensional quantum spin systems at finite temperature. Using the cluster expansion, we can represent the model's thermal density operator at inverse temperature $\beta = 1/T$,

$$\rho(\beta) = \frac{1}{\mathcal{Z}} e^{-\beta H}, \quad \mathcal{Z}(\beta) = \text{Tr}(e^{-\beta H}), \tag{47}$$

as a tensor network operator of the form

$$\rho(\beta) = \quad . \tag{48}$$

Here, the bond dimension $D$ of this tensor network operator is determined by the order of the cluster expansion; we expect that the approximation becomes better as we increase $D$. Note

that obtaining this tensor network comes at negligible numerical cost.

The partition function $\mathcal{Z}$ is then obtained by tracing over the physical degrees of freedom, such that we obtain a simple two-dimensional tensor network

$$\mathcal{Z}(\beta) = \qquad\qquad . \tag{49}$$

This tensor network can be efficiently contracted using standard methods such as the variational uniform MPS (VUMPS) algorithm [15–17], the corner transfer matrix renormalization group (CTMRG) [18–20] or real-space renormalization-group approaches [21, 22]; in this work, we use the first option. Here, the bond dimension $\chi$ of the boundary MPS enters as a control parameter. The leading computational complexity of both the above methods scales as $\chi^3 D^2$, where $D$ is determined by the order of the cluster expansion and the appropriate scale of $\chi$ depends on the entanglement in the system. Performing this contraction yields a direct calculation of $\lambda$, the scaling of the partition function with system size in the infinite-size limit, such that we obtain the free energy density $f$

$$\mathcal{Z}(\beta) \propto \lambda^{N_x N_y} , \qquad f(\beta) = -\log\lambda . \tag{50}$$

In addition, using the boundary MPS we have direct access to the local reduced density matrix

$$\tag{51}$$

which allows us to compute local observables directly in the thermodynamic limit.

As an illustration of the power of this method, we study the thermal phase transition in the transverse-field Ising model on a square lattice, defined by the Hamiltonian

$$H = -\sum_{\langle ij \rangle} \sigma_i^x \sigma_j^x + g \sum_i \sigma_i^z . \tag{52}$$

The thermal phase diagram is plotted in Fig. 5, showing a line of thermal second-order phase transitions between an ordered ferromagnetic phase and a disordered paramagnetic phase. In the classical limit ($g = 0$) there is the phase transition of the classical Ising model at $\beta = \log(1 + \sqrt{2})/2 \approx 0.44$, whereas in the zero-temperature limit ($\beta \to \infty$) we find a quantum phase transition at $g \approx 3.044$. For any non-zero temperature, the phase transition falls within the 2-D classical Ising universality class.

First we focus on the phase transition at a fixed value of the field, $g = 2.5$. We represent the density operator as a tensor network operator of dimension $D = 27$ by a cluster expansion of order five and loop correction. In Fig. 6 we plot the results from VUMPS simulations at different values of $\chi$. First we plot the magnetization as a function of $T$ that we have obtained for different values of $\chi$. We clearly see the Ising phase transition, but the critical point is shifted due to finite-$\chi$ effects. In order to get an accurate simulation of the phase transition, we

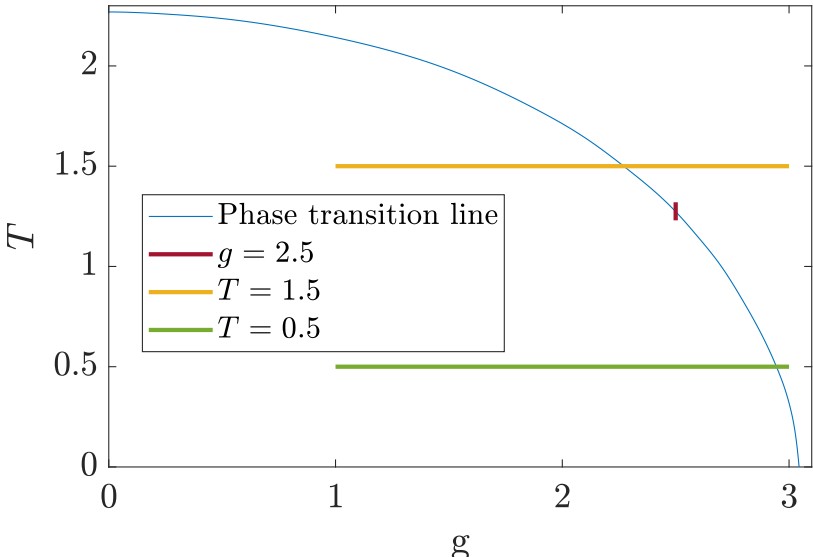

Figure 5: The phase diagram of the 2- transverse-field Ising model: the ordered ferromagnetic phase is separated from the disordered paramagnetic phase by a second-order phase transition. We show the lines of constant $T = 2.5$, $g = 0.5$ and $g = 1.5$ that are used in Figs. 6 and 7. The data for the transition line is taken from the Monte-Carlo data in Ref. [23].

can employ a finite-entanglement scaling approach [24]: We extract an effective length scale $\delta$ from the spectrum of the boundary-MPS transfer matrix, and apply the scaling hypothesis

$$\tilde{m}\left(t\delta^{-1/\nu}\right) = m(t,\delta)\delta^{-\beta/\nu}, \tag{53}$$

with $t = T - T_c$ and $(\beta, \nu)$ the known Ising critical exponents. We can optimize the value of $T_c$ such that we get a good collapse of the scaling function $\tilde{m}$ [24]. The optimized data collapse is plotted in (b), where we have obtained a value of the critical temperature of $T_c = 1.2736(0)$. This value should be compared to the quantum Monte-Carlo estimate $T_c = 1.2737(6)$ [23] and a PEPS estimate $T_c = 1.2737(2)$ [25]. This good agreement for $T_c$ illustrates the fact that our fifth-order cluster-TNO is an extremely good approximation of the partition function. Finally, in the right panel of Fig. 6 we also show the data collapse for the correlation lengths that we extract from the boundary MPS, using a similar scaling hypothesis [24]

$$\tilde{\xi}\left(t\delta^{-1/\nu}\right) = \xi(t,\delta)\delta. \tag{54}$$

Again, we find a collapse of the data.

Note that the PEPS method [25] must contract the square of the partition function ($\rho^\dagger\rho$), making the contraction much more costly. Additionally, one has to optimize the tensors first, which has a non-negligible numerical cost as well.

Of course, the truncated cluster expansion is expected to break down when decreasing the temperature. In order to illustrate this, we have simulated the phase transition for two fixed values of the temperatures, and for different orders of the cluster expansion. In Fig. 7 we plot the magnetization as a function of the field, again showing that the Ising criticality is always found, but where the value of the critical field is shifted. For $T = 1.5$ we find that the critical point approaches the exact value to a high precision, whereas for $T = 0.5$ the order-six cluster-TNO yields a value of the critical point that is significantly shifted with respect to the exact value.

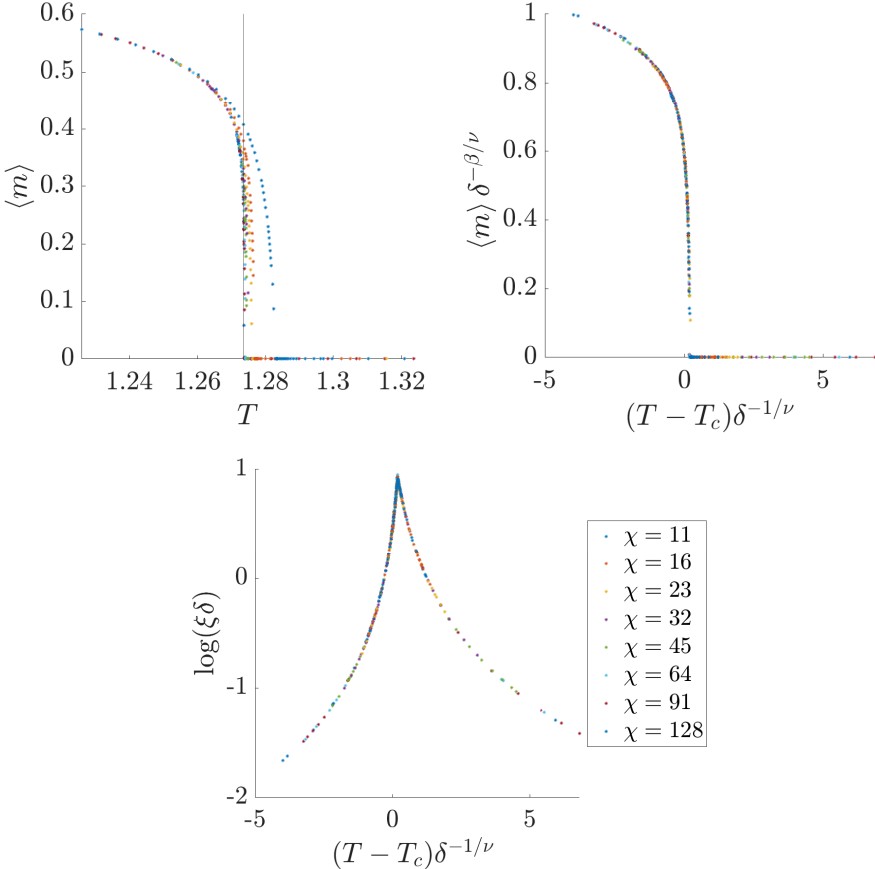

Figure 6: Results for 2-D transverse-field Ising model at $g = 2.5$, obtained by the VUMPS contraction of an order-six cluster TNO. The left panel shows a direct calculation of the magnetization as a function of $T$ for different values of $\chi$. The middle panel shows the rescaled magnetization, according to the scaling hypothesis, where we have optimized $T_c$ to yield the best collapse of the data. The right panel shows the rescaled data for the correlation length extracted from the boundary MPS. The data set is sampled at values of $\chi = 11, 16, 23, 32, 45, 64, 91, 128$ with roughly 65 data points for each value of $\chi$.

To study this regime with tensor networks, one might use several layers of cluster expansion TNO's, the PEPS method referred to earlier, or indeed some optimal hybrid of the two.

# 5 Outlook

In this paper, we have explained how to represent cluster expansions as tensor-network operators, which can be used efficiently in tensor-network simulations for real- and imaginary time evolution of local Hamiltonians. We have shown that the cluster-TNO construction yields an extremely simple way of representing the partition function of 2-D quantum spin systems at non-zero temperature, which despite its simplicity gives accurate results for relatively low temperatures. This approach should be compared to the standard tensor-network approach, where the thermal density operator is represented as a projected entangled-pair operator, and evolved by imaginary-time evolution through a Trotter-Suzuki decomposition of the density operator [26–28].

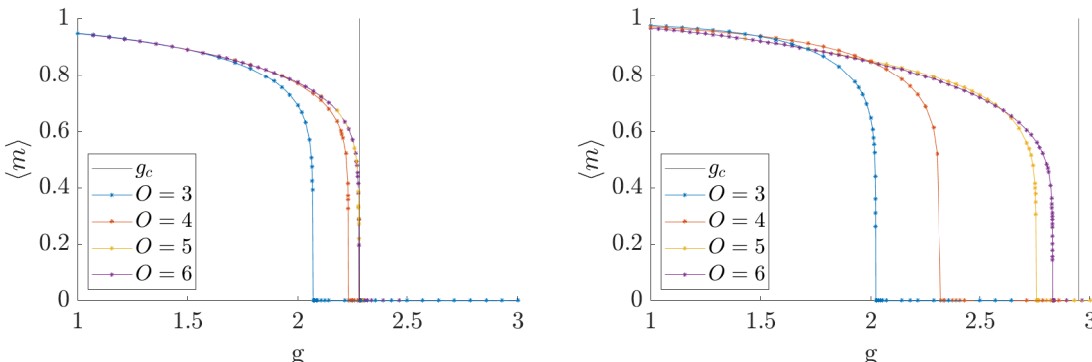

Figure 7: Results for the 2-D transverse-field Ising model at fixed $T = 1.5$ (top) and $T = 0.5$ (bottom). The magnetisation is calculated for MPS bond dimension $\chi{=}45$. The figures show the magnetisation curve for different orders $O$ of the cluster-TNO. For order $O = 6$, virtual level '3' is truncated at bond dimension 20. The black lines denote the "exact" critical temperatures, taken from Ref. [23].

As our benchmarks for the 1-D case have shown, the cluster expansion breaks down for small temperatures. In that case, however, we can think of splitting up the thermal density operator into a sequence of cluster TNOs. Since the bond dimension of this TNO would grow exponentially with the number of layers, intermediate truncation steps will be necessary here – a variational truncation scheme seems to be the best option. Here, again, we believe that the cluster-TNO will be better suited than a Trotter-Suzuki decomposition of the density operator, since the former allows us to take much larger imaginary time steps.

# Acknowledgements

This work was supported by the Research Foundation Flanders and ERC grant QUTE (647905).

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
