# Peer review of "Simulating thermal density operators with cluster expansions and tensor networks"

_SciPost Physics, doi:SciPost Phys. 14, 085 (2023)_

## Round 1 · Referee Report · Anonymous (Referee 1) · 2022-10-11

Strengths

The manuscript proposes a highly original approach to thermal states of quantum Hamiltonians. In some applications it may surpass other methods based on tensor networks.

Weaknesses

What I miss in the manuscript is a more quantitative discussion how the numerical cost of performing the algorithm depends on the cluster size or the correlation length (or whatever else is more relevant). Could the main bottleneck be identified? It would help a reader to figure out where this is the method of choice and where it is not.
To be more specific, for g=2.5 the critical temperature is in agreement with a PEPS [25] and a quantum Monte Carlo [23] but references [23,25,27] also consider g=2.9 where the critical temperature is much lower but the PEPS still agrees with the QMC. What precisely prevents comparison also in this case? The same question could also be applied to the bottom panel in Fig. 7 where g_c is close g=2.9 and something prevents the same accuracy as in the top panel. What is it?

Report

This certainly is a publishable work but, given its novelty, I would like the authors to elaborate more on what sets the limits to the proposed method.

---

## Round 2 · Referee Report · Anonymous (Referee 1) · 2022-10-14

Report

I am satisfied with the response and the extra remarks in the manuscript.

---

## Round 2 · Referee Report · Didier Poilblanc (Referee 2) · 2022-12-22

Strengths

This work introduces a powerful tensor network method to deal with real/imaginary time evolution, a very timely topic.

Weaknesses

I did not find any real weakness.

Report

The field of out-of-equilibrium dynamics/thermodynamics is evolving fast. This work introduces a very efficient method to compute time evolution using tensor network, overcoming some of the limitations of previous work. I find it very promising for future applications besides the usual 1D/2D traverse fields Ising models.

Requested changes

Suggestion: may the other write down the 1D models displayed in Eqs. (40), (41) (46) explicitly as 1D model, using sum_i instead of \sum_<ij> since the whole section is devoted to 1D and besides, the 2D version of the models are introduced again in a subsequent section IV.

  • validity: high
  • significance: top
  • originality: top
  • clarity: high
  • formatting: excellent
  • grammar: perfect

Author:  Bram Vanhecke  on 2023-01-17  [id 3244]

(in reply to Report 2 by Didier Poilblanc on 2022-12-22)
Category:
answer to question

We thank Dr. Poilblanc for reviewing our paper.

We will make the desired changes to the Hamiltonian definitions.
We also agree that the old definitions could benefit from a clearer form.

Attachment:

ClusterPepo_1.pdf

---

## Round 2 · Author Response

We thank the referee for taking the time to evaluate our manuscript.

We have tried to add in all the missing details one might need to decide if this is the appropriate method for the case one might be interested in. This includes a mention of the numerical complexity in the case of 2D, as well as a more in depth comparison to the alternative tensor network approach. We also clarified that the method works wonderfully for beta/t below or around unity, yet falters above that.

---

## Round 2 · List of Changes

added a subsection 'discussion' the seciton 'benchmarks'

elaborated on the numerical complexity just above eq. 50

added comparison with PEPS method under eq. 54

included possibilities for low temperature above 'outlook'

---

## Editorial Decision

published